# Student intentions to continue with distance learning post-COVID: An empirical analysis

Adriana Aletta Steyn[1]*, Craig van Slyke[2], Geoffrey Dick[3], Hossana Twinomurinzi[4], Lateef Babatunde Amusa[4,5]

1 Department of Informatics, University of Pretoria, Pretoria, South Africa, 2 College of Business, Louisiana Tech University, Ruston, Louisiana, United States of America, 3 St John's University, Queens, New York, United States of America, 4 Centre for Applied Data Science, University of Johannesburg, Johannesburg, South Africa, 5 Department of Statistics, University of Ilorin, Ilorin, Nigeria

☯ These authors contributed equally to this work.
* riana.steyn@up.ac.za

## Abstract

The aftermath of COVID changed how students learn, mainly moving to a distance learning model. The research reported in this paper investigated the organizational and individual factors that influence the preference for continuing with distance / online learning post-COVID. Partial least squares structural equation modeling was applied to a model developed for this research, based on data from 452 students from residential universities in South Africa. The key results reveal an overall reluctance to continue with distance learning. This is despite the technological and faculty support offered to university students and how distance learning fits their learning styles. This is likely due to student living conditions and their perception of low institutional concern. On the other hand, faculty support has a more substantial impact on continuance, compared with a generally negative perception of support from the universities. The research underscores the importance of addressing student reluctance to continue with distance learning by improving institutional support and tailoring learning styles. The research enhances our understanding of crucial factors influencing students' preference for distance/online learning post-COVID. It also underscores the pedagogical shifts brought about by the pandemic, particularly highlighting the changing roles of faculty support and the impact on students' living conditions.

## Introduction

The 'Severe Acute Respiratory Syndrome Coronavirus 2' (SARS-CoV-2), now known as Corona, or Covid-19, has created havoc across global communities, forcing countries to implement widespread measures to combat the spread of the virus. One of the more drastic measures included "lockdown," causing the restriction and even closure of many economies, schools, and universities [1–5]. This assisted in accelerating the endless call to move away from only traditional learning models- a rebirth [6] if you may, as Covid-19 sped up this transition [5] for many universities. Kandri [6] argues that of all the educational "levels" the higher education environment will "trigger a learning revolution", this is echoed by Kativhu [7], stating a

**Data Availability Statement:** All relevant data are available from the figshare repository (https://figshare.com/s/e787e7d3c9d7c181f76d).

**Funding:** The author(s) received no specific funding for this work

**Competing interests:** The authors have declared that no competing interests exist.

long overdue need for digital learning adoption. This is due to the type of student who is tech savvy enough to easily "convert" to a new technology platform but also old enough to go entirely online with endlessly demanding content. Because of Covid, and undoubtedly long after Covid-19, this "appetite" for online learning has and will continue to grow [6]. On the flip side, though, Van Slyke, Clary [4] and Clary, Dick [8] noted that some students prefer face-to-face learning as they signed up for it. The questions raised in those papers remain: what will encourage students to continue with online learning now that the pandemic seems to be ending, and what will affect that intention?

The 2022 trends in higher education suggest that this need continues to increase, with the top 3 trends being 1. "Students will expect convenience and flexibility"; 2. "You'll Need Higher-Quality Online Courses" and lastly, "Overlap Between Traditional Degree-Based Learning and Skills-Based Learning" [9]. The concept of quality is also echoed by Weldon, Ma [2], who identified the need for quality, comprehensive, and well-designed online material, and assessments. However, this takes time to develop in an ideal world, something we did not have during Covid.

As universities, students, and academics emerge from the pandemic, they face many problems in continuing with online classes.

Classes are being retained, modified, or discontinued—now is the time for comprehensive and wide-ranging studies of potential approaches and achievements. As they face these problems, administrators must draw on reliable data for guidance. The authors believe that one of the most important factors affecting these considerations is what the students think. This study, focusing on student intentions, aims to contribute to administrators' understanding of what needs to be done.

Previously, many authors have called for educators to understand better our students' connection to the digital world [10–12]. In this context, Seemiller and Grace [13] noted that Generation Z students prefer independent learning, where engaging with educators is perceived only as a valuable resource when and if needed. A study conducted in Hong Kong showed that because of being forced into an online learning environment, online learning actually supported students to become independent learners and to develop critical skills, but more importantly, it improve peer collaboration across various subjects [2]. Collaboration and avoiding isolation was a huge concern for many during the pandemic, as they wanted to feel connected to others. Yet, a study by Weldon, Ma [2] mentioned increased collaboration.

Although Zheng, Lin [1] noted that a feeling of being isolated brings on loneliness and is even associated with feelings of social pain, being socially connected is, in fact, fundamental to our survival as humans. A lack of human interaction was also noted as a byproduct of the online learning environment, particularly when specifically moved from a prior face-to-face approach [2]. They continue this theme to suggest implementing instant messaging and video conferencing more effectively to combat this problem. However, A benefit of online learning was the ability (provided by the pandemic) for academics and students to have a more flexible schedule [2]. This allowed educators to integrate their work/life better, thus more effectively adapting to the added workload the pandemic added to these educators. But what happens when we must adjust to a new way of learning when students and educators must attend contact classes again? How will this look? What will students see? What do students want? What support will institutions and faculty need to provide for a smooth transition? This paper investigates student satisfaction and the desire to continue with online learning in residential universities (institutions where most students live on campus) in South Africa. It aims to identify those factors that will affect the future possible take-up of online education.

While many such studies have been conducted in the past, the pandemic may have modified attitudes and practices. The authors believe it is also the first such study in South Africa.

Kativhu [7] raised a concern about the adoption of online learning in Africa; only 24% have access to the Internet. Although South Africa's internet adoption is a bit more attractive (nearly 80% internet usage [14]), the fact that South Africa also has an energy crisis means that access to power as well as the Internet in more remote and rural areas, in South Africa, make online learning less attractive to students [7, 15]. Kativhu [7] also continues to note that online learning in developing countries, such as South Africa, and particularly rural areas, is complicated by factors such as the lack of educational infrastructure, human resources, or even previously disadvantaged communities, not to mention the current energy crisis as well [15]. Similarly, Makombe [15] noted that due to the pandemic, the notion of academic exclusion started to rise again, particularly due to some students' poor socio-economic circumstances, especially in townships, informal settlements, and rural areas.

However, there is a flip side, the support provided during the pandemic was a temporary technology adoption steppingstone for higher education institutions [7]. For example, during the pandemic, mobile data service providers partnered with higher education institutions [7], providing free access to learning management systems at many South African universities. This partnership, however, ended once the lockdown regulations started to ease up, and thus some students, again, needed access or sufficient infrastructure at home once these agreements stopped.

The above leads to the following research question:

- What factors influence students' desire to continue with distance learning after the circumstances that required it no longer prevail?

The study contributes to the literature about the post-pandemic world by providing insights into the factors influencing students' intentions to continue with distance learning post-COVID, shedding light on individual and organizational dynamics. By exploring the intricate interaction of these factors, the study enhances our understanding of the pedagogic shifts brought about by the pandemic. By disentangling the subjective nuances of student preference and the organizational structures that facilitate or impede distance learning, the study can potentially guide educational institutions and policymakers in their strategic decision-making process.

The remainder of the paper is structured as follows: the next section reviews the literature on institutional and individual factors that influence learning, leading to a model to test. We have placed the model early in the section to aid understanding, followed by its substantiation. The methodology and the results follow it. There is a discussion section, and the paper ends with conclusions and areas for further research.

## Literature review and theoretical development

The education sector around the world was significantly affected by the Covid-19 pandemic. Almost overnight, millions of students in higher education were sent home, away from their instructors and sometimes into a lesser technological environment [15]. A recent UNESCO Report has noted that the rapid return to in-person teaching suggests that while a fundamental transformation of the higher education scene is not likely, many institutions are likely to take some of the digital practices into the longer term [16]. The report also states:

> "*The impact of the pandemic in higher education has been diverse, profound and different for each institution and country.*"

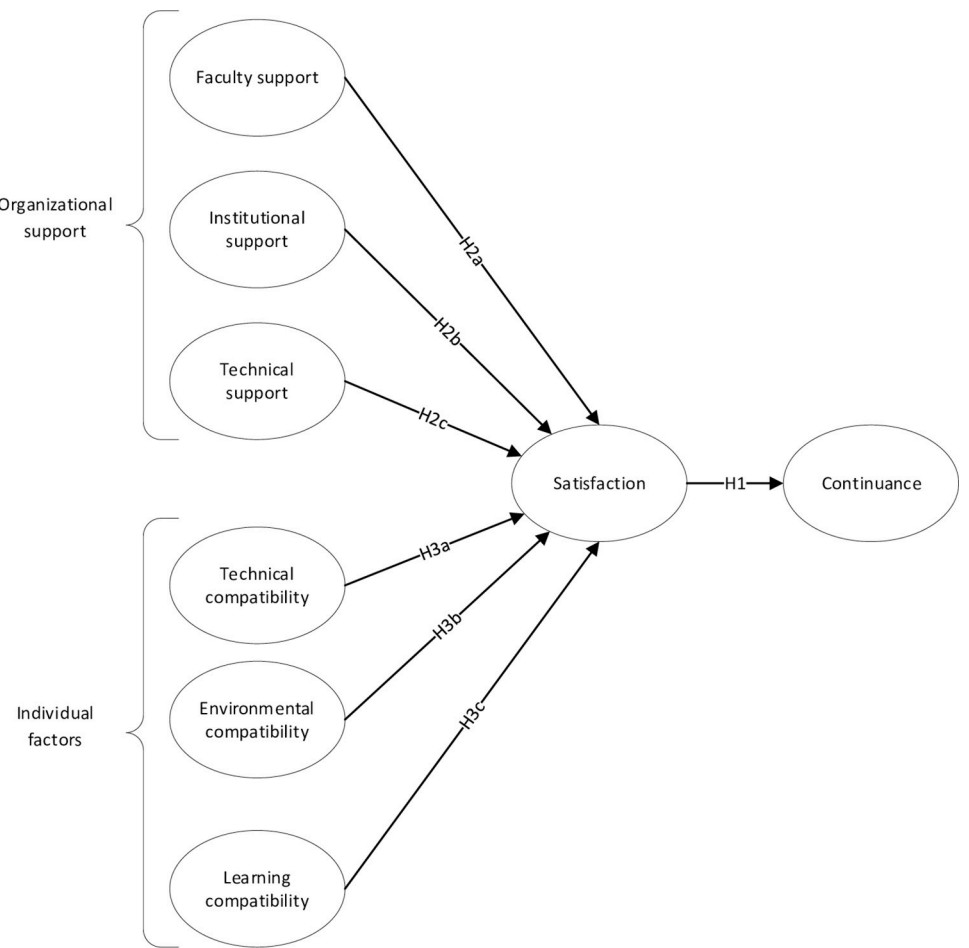

**Fig 1. Theoretical model to test continuance with online learning (Source: Authors).**

Accordingly, reviewing the individual components that influence distance learning at the individual country and institutional level is important. The experiences of all concerned in higher education over the last 2 or 3 years have no doubt affected the desire to continue with this form of instruction, either positively or negatively. Using Social Cognitive Theory and Social Cognitive Career Theory, satisfaction with distance learning has been shown to have a strong positive effect on the desire to continue with distance learning [8], and that Support and Compatibility factors largely drove this satisfaction.

Fig 1 below represents the research model we used in our study. We expect the Support and Compatibility factors to affect student satisfaction independently and cumulatively with the online learning environment which we, in turn, expect to influence Continuance.

## Distance learning satisfaction

Distance learning satisfaction is the extent to which students find distance learning to be a pleasurable, rewarding experience (adapted from [17]). The concept of satisfaction has been applied to many domains (e.g., job satisfaction, consumer satisfaction), including distance learning [18–20]. In a largely mandatory environment, such as the use of distance learning during COVID-19, satisfaction may not be a strong predictor of behavior. However, in volitional environments, satisfaction has been linked to behavioral intentions and outcomes in

several domains, including employment [21], consumer behaviors, [22], and distance learning–the perceptions of students related to learning compatibility influenced their desire to continue with distance learning through satisfaction [8].

As COVID-19 restrictions ease, satisfaction with distance learning may be an important determinant of intentions to continue with distance learning. For many students the COVID-19 restrictions would have provided their first exposure to this form of instruction–along with the associated advantages and challenges [23]. There is empirical evidence of a strong relationship between distance learning satisfaction and continuance intentions (e.g. [8, 20]). Students who are dissatisfied with distance learning are likely to return to face-to-face courses when they can. The opposite is also true; satisfied students should be more inclined to continue to engage in distance learning. This response is related to the degree with which the product appeals to the consumer, in this instance, the extent to which full-time residential students found the experience of distance learning during COVID a pleasurable experience. We particularly argue that students who are satisfied with distance learning are more likely to continue with it. For these reasons, we expect that distance learning satisfaction will affect continuance intentions, as hypothesized below:

H1: Students' satisfaction with distance learning will positively influence their intentions to continue with distance learning.

## Support

Support is important for students, especially when dealing with unexpected, abrupt changes in their educational experiences. Such support is an important aspect of the educational environment and has an effect on students' satisfaction with distance learning [8, 24]. Support has also been shown to be important in other aspects of education, including choice of career and educational paths [25], and retention [26].

Organizational support theory indicates that members of organizations develop global beliefs regarding the extent to which the organization is concerned about them and their well-being [27]. The theory furthermore focuses on the employee perception of the extent to which an organization cares about their well-being and their contributions. These beliefs are associated with numerous organizational outcomes, including absenteeism, commitment, attachment, and achievements, among others. This similarly relates to students, the extent to which they perceive that their organizations are concerned about their well-being, and their contribution to the university [28]. For example, students who feel that they are not well supported may lose interest and withdraw from their studies.

Social exchange theory serves as the theoretical lens for organizational support theory [28]. Essentially, social exchange theory indicates individuals view social interactions implicitly using a cost-benefit perspective when deciding whether to pursue, advance, or terminate social relationships [29]. When an individual feels that social efforts made towards another are appropriately rewarded, those efforts and associated behaviors are reinforced. The opposite is also true, when one feels that their relationship efforts are not rewarded, one is less likely to continue those efforts.

With respect to organizations, when members feel valued, they are more likely to feel generous towards the organization. Members attribute human-like characteristics to the organization and treat their social interactions with its representatives as a sort of social exchange. Organizational support is seen as a signal that the organization reciprocates the member's efforts by being supportive of the member's well-being [28]. Further, organizational support is seen as a buffer for future situations in which aid is needed in dealing with stressful situations [28] such as dealing with the challenges of distance learning.

In the context of distance learning, three forms of support are seen as important: support from the institution, support from faculty, support for technical issues. Moving from the organizational environment, perceptions of institutional support is an overarching belief about the support available from non-specific others who work for and represent the institution. Faculty support beliefs are more specific; they concern the level of caring and support available from the student's professors and instructors. Technical support is specific with respect to the type of support needed. There is support for this too from studies of student acceptance of MOOS [30]. Although other dimensions of organizational support may exist, we contend that these three forms represent a useful range of support relevant to distance learning.

It is important to note that according to organizational support theory, organization members develop global beliefs regarding organizational concern for their well-being. Organizational membership is a holistic experience; it is difficult for individuals to make sharp distinctions about support for discrete aspects of the organizational member experience. Therefore, we consider global beliefs regarding the support the educational organization offers students rather than support that is specific to distance learning. This is consistent with prior studies of distance learning (e.g. [8]).

## Institutional factors

**Faculty support.**   For most students, faculty are the main connection between students and universities as institutions. Many students have limited interaction with other official university representatives such as administrators. Therefore, the extent to which faculty are seen as being supportive and caring towards students is likely to have an influence on students' satisfaction with distance learning especially when the transition to distance learning is chaotic and forced on the students. As has been recognized for some time, there is a considerable workload associated with the development and delivery of online classes [31] and now that the role of the instructor has moved to one of a facilitator [32] this workload has only increased. As noted earlier, the pandemic has led to considerable uncertainty and stress among students (and faculty). The ability of instructors to cope in the online environment is a factor too [33]. This complex interplay between uncertainty, ability and pandemic related stress was exacerbated by the dynamic nature of universities' responses to COVID-19 [34]. In some cases, the changes seemed almost constant. Such shifting sands are difficult for students to navigate, bringing about additional stress. Because of this dynamic, uncertain, stressful environment, we can surmise that faculty support will be especially important in determining the extent to which students are satisfied with their distance learning experiences, as stated in the hypothesis below:

H2a: The perceived faculty support will positively influence students' satisfaction with distance learning.

**Perceived institutional support.**   In the context of distance learning, support from the institution is an important factor in whether students find learning online to be a satisfying experience. Akbulut-Bailey [35] showed that such support not only improved learning but encouraged students to follow particular academic paths. Ho, Cheong [36] found that institutional support improved student performance (measured in dropout and retention rates). Institutional support was identified as an important factor in retention by [37]. Distance learning is a relatively new experience for most students. Due to the lack of experience, students face considerable uncertainty with respect to how to successfully navigate the challenges of learning online. Although the technical mediation between the student and the institution is

an important source of uncertainty, the entire experience is similarly uncertain. For example, students who have questions or problems may be used to talking with instructors after class or may drop by the instructor's office. These informal avenues of interaction are not available in distance learning. During active learning activities such as labs, students may be used to simply raising their hands when they need help. Although some distance learning systems attempt to simulate hand-raising, the mechanisms are somewhat cumbersome and unfamiliar to most students.

These uncertainties go beyond the classroom. Students may also struggle to substitute informal social interactions with their peers in an online modality. Similarly, they are unable to drop by department offices when they need administrative help. So, students face multiple uncertainties, all of which are likely to result in some level of stress [4]. Because of this, the university's support and concern for student well-being is an important factor in determining the level of satisfaction students have with distance learning. This logic is stated in the hypothesis below:

H2b: The perceived institutional support will positively influence students' satisfaction with distance learning.

**Technical support.** Information technology is an important enabler for distance learning. Students sometimes struggle with technological difficulties when adjusting to distance learning, especially when distance learning is mandated to students who are expected to be taking face-to-face classes. As online education is increasingly seen as asynchronous (read "anywhere, anytime") round-the-clock technical support is particularly important in backing such classes [38]. Because of the central role of technology, students who encounter technology difficulties are likely to experience significant distress, which activates the awareness of organizational support related to addressing such problems. Readily available, competent technical support is an important signal of the extent to which the university cares about students and their well-being. This, however, was not included in the model as the focus was on technical support and not necessarily immediate availability or competency. The perception that online learning is being well delivered is shown to be highly important in determining the satisfaction scores [36]; support for technical problems is likely to be a component of well-delivered distance learning. When students feel that they have adequate technical support from the university, they are likely to feel more favorable towards distance learning, ("someone cares") and therefore will be more satisfied. This logic is reflected in the following hypothesis:

H2c: The perceived technical support will positively influence students' satisfaction with distance learning.

## Individual factors–Perceived compatibility

We drew on the diffusion of innovation theory to make sense of how individuals adopted distance learning [39]. In Rogers [40] innovation-decision process model, beliefs about the characteristics of an innovation are important to the decision to adopt or reject an innovation. In the context of distance learning, compatibility has been shown to be important to intentions to continue or discontinue online learning [8, 41], as well as satisfaction with distance learning [8]. In fact, Clary et al. [8] found that compatibility had the strongest effect on satisfaction among the variables included in their study. These authors included three forms of compatibility, compatibility with learning styles, technical compatibility, and environmental compatibility.

**Learning style compatibility.** Compatibility pertains to the adopter's existing values, previous experiences, and needs. Compatibility links to the innovation tying into the adopter's situation leading to a sense of familiarity.

In the context of distance learning, compatibility with students' preferred style of learning is important. Note that we are not referring to Kolb's learning styles [42–44] and similar frameworks. Rather we are equating learning style compatibility to work style compatibility [45] which reflect a student's preferred way of engaging in their learning activities. Attitude and self efficacy were key factors driving acceptance of blended learning in s study by [33]. This is similar to the way compatibility has been portrayed in other studies of distance learning compatibility (e.g., [46, 47]). Students who believe that distance learning fits with their preferred way of learning are more likely to be satisfied with distance learning, as stated in the following hypothesis:

H3a: The perceived learning compatibility with distance learning will positively influence satisfaction with distance learning

**Technical compatibility.** During COVID-19 restrictions, students were forced to learn online regardless of the extent to which they found the situation compatible with their preferences and situation [1–4, 48]. As a technology-mediated modality, students' satisfaction with the distance learning experiences are likely shaped, in part, by the extent to which the use of technology is compatible with their situation. There are at least two important aspects to this, the extent to which students are comfortable with using learning-related technologies to support their learning, and the availability of the required technologies. Positive beliefs related to technical compatibility should result in lower distance learning related stress and therefore higher satisfaction. When the use of technology for coursework is incompatible, students are likely to experience stress and other negative affective reactions as they attempt to deal with the problems related to low technical compatibility. This will, in turn, affect their satisfaction with distance learning, as reflected below:

H3b: The perceived technical compatibility with distance learning will positively influence satisfaction with distance learning.

**Environmental compatibility.** During the response to COVID-19, many universities effectively shut down most of their physical operations, typically including closing campuses and student housing facilities. For many students, this shutdown meant a return to their parent's home. Often students were forced to share space with siblings and parents who were trying to learn and work from home. Because of this, many students struggled to find suitable spaces for learning [8, 49]. Students who found their environments incompatible with learning online were subject to additional stress beyond that caused by the transition to distance learning. Coping with environmental challenges likely sapped the psychological and mental energy for many of these students, leading them to be less favorable towards distance learning. We expect that this will result in reduced satisfaction. H2c, which is presented below, reflects this thinking.

H3c: The perceived environmental compatibility with distance learning will positively influence satisfaction with distance learning.

## Research method

The model depicted in Fig 1 above was tested by a cross-sectional survey administered to students at two large South African universities, following a quantitative approach in obtaining

the data, with appropriate institutional approval. The data was collected in mid-2022. Scales were drawn from previous published studies. In some cases, scale items were adapted to the context of our study. With the exception of scales related to support (faculty support, technical support, and institutional support) items were adapted to the context of distance learning. The wordings for the support scales are more general; we left these general because it is difficult for students to separate general support from support that is specific for distance learning. For example, when evaluating whether their university is supportive, it is difficult for students to parse out distance learning and other aspects of their university experience. So, we worded the support items to be more general. Scale items and their sources are provided in S1 Appendix.

## Data collection

The target population was college students in South Africa. Due to the unavailability of a sampling frame, a non-probabilistic sampling technique, convenience sampling, was utilized. This study performed a preliminary power analysis using the using a web power online tool [50] to determine the minimal sample size. It was then determined that 444 respondents were adequate for the current study. Overall, the data collection provided 453 responses, exceeding the previously indicated minimal sample size criteria.

## Common method bias

Due to our data's cross-sectional and self-reported nature, common method bias is possible. Our study is vulnerable to the inflation of correlations by common method variance (CMV). As an apriori approach to controlling CMV, the order of the scale endpoints was varied. Further, a data validation test, the full collinearity assessment of Kock [51], was employed. The variance inflation factor (VIF) values at the factor level were assessed after running the consistent PLS algorithm for each construct (one at a time) being assumed as the dependent variable. All the constructs had VIF values below 3, confirming that the model was free of common method bias. We also conducted a marker variable analysis [52] using the attitude towards the color blue scale [53]. The mean correlation between the marker variable and the other latent variables was 0.06, which is less than 0.10, indicating that CMV is not a serious problem [54].

## Data analysis and results

Our analytic sample comprised 453 respondents. As can be seen in Table 1, males and females made up 42% and 57% of the total population, respectively. The mean age of the respondents was 20.6 years.

We used partial least squares structural equation modeling (PLS-SEM) from the SmartPLS 3.3.5 software [41] to empirically examine our theoretical framework. The PLS-SEM technique is a causal modeling method that aims to maximize the variance of the dependent latent constructs explained by the independent variables. The rationale for using PLS-SEM (variance-

**Table 1. Demographic profile of respondents (n = 452).**

| Variables | N | % |
|---|---:|---:|
| *Gender* | | |
| Male | 191 | 42.3 |
| Female | 258 | 57.1 |
| Other/ not answered | 4 | 0.7 |
| | **Mean** | **SD** |
| Age (years) | 20.6 | 3.2 |

**Table 2. Scale descriptive statistics.**

| Variable | Mean | Standard Deviation | Skewness | Kurtosis |
|---|---|---|---|---|
| Continuance intention | 3.698 | 2.171 | 0.138 | -1.445 |
| Environmental compatibility | 5.195 | 1.471 | -0.778 | 0.037 |
| Faculty support | 2.968 | 1.222 | 0.717 | 0.720 |
| Institutional support | 4.664 | 1.389 | -0.456 | -0.211 |
| Learning compatibility | 4.139 | 2.078 | -0.074 | -1.343 |
| Satisfaction | 3.572 | 1.026 | 0.365 | 0.210 |
| Technical compatibility | 2.550 | 1.223 | 1.012 | 0.989 |
| Technical support | 2.706 | 1.270 | 0.791 | 0.600 |

Note: Descriptive statistics were computer using Python version 3.8.10 through GPT4's Code Interpreter.

based SEM) in this study arises from two unique factors. First, because this study focuses on the factors influencing mobile distance learning rather than its confirmation, PLS-SEM is seen as a more appealing and acceptable technique in such circumstances [50]. Second, we are aware of PLS-SEM's relative robustness to complicated structural models with numerous constructs, indicators, and model relationships [55].

The measurement model evaluated the constructs' internal consistency, reliability and validity, while the structural model tested the hypothesized associations between constructs. Where necessary, we used the average estimates from bootstrapped 5000 subsamples to assess the significance of the parameter estimates [56].

Descriptive statistics for each of the hypothesized variables is shown in Table 2. There are deviations from normality, as indicated by the skewness and kurtosis statistics, however our analytical method, PLS-SEM is robust with respect to non-normality.

## Measurement model

The assessment of the measurement models includes internal consistency (Cronbach's alpha and composite reliability), convergent and discriminant validity. As shown in Table 3, the Cronbach's alpha and composite reliability values were above 0.7, reflecting internal consistency reliability. We assessed convergent validity through indicator reliability and average variance extracted (AVE). In terms of indicator reliability, the indicator's standardized outer loadings were above the critical threshold of 0.7. The AVE values were well above the 0.5 threshold, supporting convergent validity [57].

Table 4 shows the results of the heterotrait-monotrait ratio (HTMT) assessment for discriminant validity, which is superior to both the Fornell and Larcker criterion and the assessment of cross-loadings [58]. With one minor exception, all the HTMT values were lower than the conservative threshold of 0.85. The HTMT value for work style compatibility and satisfaction was 0.867. Although this is above the strict 0.85 threshold, it is below another, well-established heuristic, which indicates that values below 0.90 are acceptable [58]. Further, a bootstrapping procedure was conducted with 5,000 samples using the bias-corrected and accelerated (BCa) bootstrap confidence interval [59, 60], showing that all HTMT values were statistically different from 1. These results provide support for discriminating validity for all constructs. We also examined the relationships between latent variable correlations and the square root of the average variance explained. In all cases, the maximum correlation for each latent variable was less than the corresponding square root of the average variance explained, further demonstrating discriminant validity [61].

**Table 3. Measurement model evaluation: Internal consistency and convergent validity.**

| Construct | Indicators | Loadings | Cronbach's α | CR | AVE |
|---|---|---:|---:|---:|---:|
| Continuance intention | CI1 | 0.969 | 0.967 | 0.969 | 0.939 |
| | CI2 | 0.973 | | | |
| | CI3 | 0.965 | | | |
| Environmental compatibility | EC1 | 0.869 | 0.841 | 0.860 | 0.756 |
| | EC2 | 0.861 | | | |
| | EC3 | 0.878 | | | |
| Faculty support | FS1 | 0.844 | 0.887 | 0.888 | 0.748 |
| | FS2 | 0.905 | | | |
| | FS3 | 0.882 | | | |
| | FS4 | 0.828 | | | |
| Institutional support | InstSup1 | 0.739 | 0.845 | 0.886 | 0.767 |
| | InstSup2 | 0.945 | | | |
| | InstSup3 | 0.928 | | | |
| Learning compatibility | LC1 | 0.963 | 0.980 | 0.980 | 0.943 |
| | LC2 | 0.963 | | | |
| | LC3 | 0.982 | | | |
| | LC4 | 0.977 | | | |
| Satisfaction | Sat1 | 0.913 | 0.932 | 0.939 | 0.832 |
| | Sat2 | 0.861 | | | |
| | Sat3 | 0.935 | | | |
| | Sat4 | 0.937 | | | |
| Technical compatibility | TC1 | 0.882 | 0.850 | 0.892 | 0.689 |
| | TC2 | 0.900 | | | |
| | TC3 | 0.795 | | | |
| | TC4 | 0.732 | | | |
| Technical support | TS1 | 0.906 | 0.897 | 0.899 | 0.830 |
| | TS2 | 0.945 | | | |
| | TS3 | 0.881 | | | |

**Abbreviations:** CR = Composite reliability; AVE = Average variance extracted

## Structural model

To assess the structural model, we followed the guidelines of Hair Jr, Hult [56] by evaluating the size and significance of path coefficients and the model's explanatory power. In addition, collinearity assessment via inner VIF values, predictive relevance ($Q^2$), and effect sizes ($f^2$) were reported. We included three control variables, age, gender, and distance learning experience (number of prior online courses taken).

The maximum latent variable VIF value was 2.049 (satisfaction and institutional support), which is below the threshold of 5 [55], suggesting no problematic multicollinearity. In terms of the model's explanatory power and in-sample fit, we examined the adjusted coefficient of determination ($R^2_{adj}$). The model reflected adequate explanatory power for the two endogenous constructs; $R^2_{adj}$ for satisfaction was 0.769 and 0.594 for continuance intentions. The $Q^2$ values for both endogenous constructs were well above zero (0.585 for continuance and 0.764 for satisfaction), reflecting the model's superiority over a naive out-of-sample prediction and an acceptable predictive relevance.

The path analytic results indicate general support for the research model, with a one exception (Table 4). Two of the support factors were significant. Faculty support (β = 0.157 p <

**Table 4. Discriminant validity assessment using Heterotrait-Monotrait ratio (HTMT) criterion.**

| | 1 | 2 | 3 | 4 | 5 | 6 | 7 |
|---|---|---|---|---|---|---|---|
| 1 | | | | | | | |
| 2 | 0.797 | | | | | | |
| 3 | 0.251 | 0.547 | | | | | |
| 4 | 0.166 | 0.370 | 0.592 | | | | |
| 5 | 0.284 | 0.515 | 0.725 | 0.694 | | | |
| 6 | 0.323 | 0.474 | 0.309 | 0.335 | 0.198 | | |
| 7 | 0.468 | 0.576 | 0.371 | 0.344 | 0.389 | 0.595 | |
| 8 | 0.831 | 0.867 | 0.341 | 0.222 | 0.307 | 0.466 | 0.512 |

Notes: 1. Continuance, 2. Satisfaction, 3. Faculty support, 4. Technical support, 5. Institutional support, 6. Technical compatibility, 7. Environmental compatibility, 8. Work style compatibility.

.0001) and technical support ($\beta$ = 0.133 p < 0.001), Fig 2, have significant relationships with satisfaction, but institutional support does not ($\beta$ = -0.004, p = 0.902). Similarly, two of the three types of compatibility had significant relationships with satisfaction, with learning compatibility ($\beta$ = 0.676, p < 0.001) having a notably stronger relationship than environmental

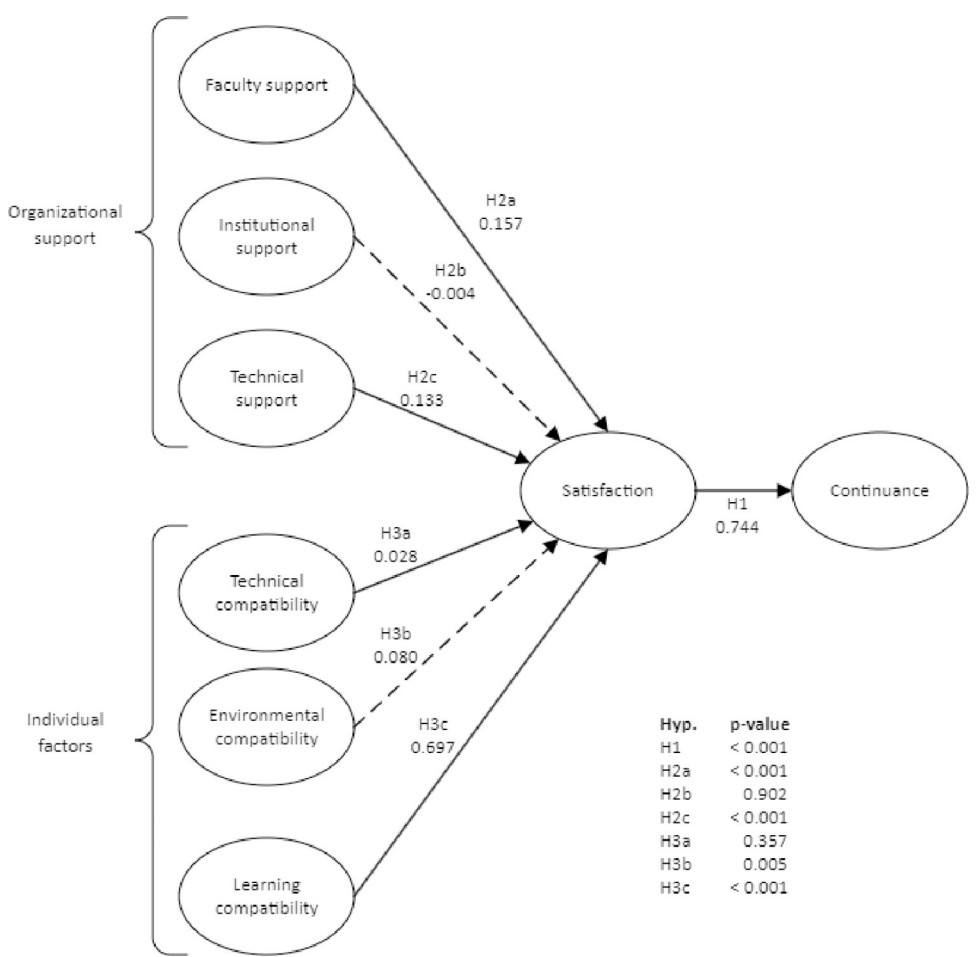

**Fig 2. Structural model (Source: Authors).**

**Table 5. Structural model results.**

| Hypothesis | Path coefficient | p-value | $f^2$ | Supported? |
|---|---|---|---|---|
| H1: Satisfaction -> Continuance | 0.744 | < 0.001 | 1.319 | Yes |
| H2a: Faculty support -> Satisfaction | 0.157 | < 0.001 | 0.058 | Yes |
| H2b: Technical support -> Satisfaction | -0.004 | 0.902 | 0.000 | Yes |
| H2c: Institutional support -> Satisfaction | 0.133 | < 0.001 | 0.038 | No |
| H3a: Technical compatibility -> Satisfaction | 0.028 | 0.357 | 0.002 | Yes |
| H3b: Environmental compatibility -> Satisfaction | 0.080 | 0.005 | 0.018 | Yes |
| H3c: Learning compatibility -> Satisfaction | 0.697 | < 0.001 | 1.475 | Yes |
| Control variables | | | | |
| Age -> Continuance | 0.101 | 0.002 | 0.024 | N/A |
| Sex -> Continuance | -0.017 | 0.589 | 0.001 | N/A |
| Distance learning experience -> Continuance | -0.041 | 0.037 | 0.037 | N/A |

compatibility ($\beta$ = 0.0780 p = 0.005); and technical compatibility was not significant ($\beta$ = 0.028, p = 0.357).

Next, we assessed effect size ($f^2$), which measures changes in an individual exogenous constructs' contribution to each $R^2$ value. As shown in Table 5, effect size magnitudes included none, small effects, and large effects. For example, satisfaction had a large effect (> 0.35) on continuance, and learning compatibility had a large effect on satisfaction. In contrast, faculty support had a small effect (between 0.02 and 0.15) while technical support, institutional support, technical compatibility, environmental compatibility had less than small effects on satisfaction.

## Discussion

Satisfaction with distance learning in residential based institutions of South Africa during the COVID pandemic was positively influenced by faculty support (social and technical support), technical support (the availability of technological support to use the distance learning technology of choice), technical barriers (ease of using distance learning technologies) and work style barriers (distance learning works for the way I learn).

Our results indicate general support for the research model. Further, the results lend credence to the use of organizational support theory and diffusion theory for studying distance learning satisfaction and continuance intentions. Our empirical findings also clearly indicate the strong influence of satisfaction on continuance intentions. Students who are satisfied with distance learning are much more likely to want to continue learning online when it is no longer mandatory. Our findings are similar to Clary, Dick [8] in that compatibility had strong effects on satisfaction and continuance intentions (desire to continue) although they found no significant relationship between support (which they modeled as a second-order factor) and satisfaction. They are also reflected to some extent in a contemporary study by [62] which reported that self directed learning had a negative effect on satisfaction.

The overarching message from our study is that both support and compatibility matter, but compatibility matters much more than support. Based on our findings, the main driver of whether students are satisfied with distance learning is the extent to which they believe distance learning is compatible with the way in which they prefer to learn. Because satisfaction had such a strong effect on continuance intentions, by extension we can say that learning compatibility is a strong, meaningful indicator of continuance intentions. Students who believe that distance learning fits with their learning preferences are substantially more likely to continue with distance learning than those with low learning compatibility beliefs. The extent to

which students believed that they had a suitable environment in which to engage in learning activities also affects satisfaction, but not nearly as much as learning compatibility. Technical compatibility did not have a significant relationship with satisfaction. Interestingly technical support was similarly non-significant, while faculty and institutional support had significant relationships with satisfaction, so, by extension faculty and institutional support seem important to continuance intentions, but technical support does not. Interestingly a study undertaken about the same time [63] in a neighboring country, Botswana, had not dissimilar findings: performance expectancy, social influence and satisfaction predict the continuance of online learning.

We find the non-significant results to be interesting as well. Both were related to technology, neither technology support nor technical compatibility had a significant impact on satisfaction. One plausible explanation for this finding is that students were already comfortable with the student-facing technology supporting distance learning. For example, many faculty members make extensive use of learning management systems. So, it is likely that many students were already used to using technology to access course materials, interact with professors and peers, submit assignments and the like. So, technical support was not important to their satisfaction; they may not avail themselves of technical support because they do not feel that they need it. Similarly, if students are already familiar with distance learning technologies, the compatibility of these technologies may already be established. Another possible explanation for the unimportance of technical support is that students may look to their instructors for support in navigating the learning management system rather than turning to a central technical support group.

Our findings also support the inclusion of compatibility in future studies of distance learning continuance intentions. Although there are numerous studies confirming the importance of compatibility to behavioral intentions, it is often excluded from studies that use theoretical perspectives such as the technology acceptance model and the unified theory of acceptance and use of technology.

The results of this study can also inform university administrators who wish to promote distance learning. One suggestion is that institutions promote the use of learning technologies such as learning management systems, online tutorials, and other tools for face-to-face courses. This suggestion mirrors work by [64] which examined the intention to use virtual meeting platforms during and after the pandemic. Not only are these tools often effective, getting students familiar with, and used to using the technologies that support distance learning will likely boost their perceptions of learning compatibility, making them more likely to engage in distance learning in the future. In addition, administrators should not only ensure that institutional support for distance learning is in place, but they should also promote and recognize faculty support for distance learning.

This last point is especially important. Faculty who teach online and provide support for online students need to believe that their efforts are recognized and rewarded, or at least not punished. Providing support for online students is time consuming. For example, faculty often need to create and manage complex learning management system environments, respond to student questions regarding online facilities, monitor discussion boards, and perhaps create or curate online learning tools such as tutorials. All of this takes time, which takes time away from other activities such as research. If evaluation and reward systems fail to recognize this, incentives will move faculty away from supporting distance learning, to the detriment of students and the continued use of distance learning.

## Conclusions

There are several implications arising from this study. First it reinforces previous findings that student compatibility with this mode of learning is all important. Therefore, it behooves

institutions who wish to facilitate it and, potentially, take advantage of increased student interest and enrollment, to ensure that it has sufficient reliable mechanisms in place to identify the suitability of those students who wish to study in this way. Secondly, the results indicate an interesting pattern in that students found their faculty offered much greater support compared with the institution. Where the institution was seen negatively, the faculty were perceived as more caring. Anecdotally, many faculty did put in a greatly increased effort during the pandemic to try and ensure their students were able to keep up with their studies–the data collected for this study appears to give some credence to this. In this regard the findings underline the importance of faculty teaching in the online environment–reward and pedagogic systems may need to be reconfigured.

The study makes several significant contributions to the literature on distance learning. Firstly, the study extends the literature on distance learning satisfaction and continuance intentions by incorporating Organizational Support Theory and Diffusion Theory. It empirically validates the strong influence of satisfaction on continuance intentions, and nuances the role of support and compatibility in this process. Importantly, the study identifies learning compatibility as a primary driver of satisfaction, highlighting the need for future research to consider this factor more prominently. Secondly, the study also provides nuanced insights into the non-significant effects of technical support and technical compatibility, suggesting that students may already be comfortable with the technology supporting distance learning. These findings are in line with previous studies but add a unique perspective from residential-based institutions in South Africa during the COVID-19 pandemic. Thirdly, the study uncovers a unique pattern where students perceive their faculty as offering more support than the institution. This observation underscores the crucial role of faculty in online teaching, suggesting that reward and pedagogic systems may need to be reconfigured to better support this mode of learning. These insights not only validate anecdotal evidence of faculty efforts during the pandemic but also provide empirical support for the importance of faculty engagement in the success of online education.

## Limitations and further research

This study was undertaken in South Africa in 2022 and while the findings are not out of line with similar studies, there may be demographic, cultural, or temporal issues that have a bearing on our work–replication studies in other countries would add to the picture. The authors recognize that while a convenience sample (the method of data collection used in this study) has generalizability limitations, in this case we believe that the sample size was a good representation of the population of those in the relevant region most affected by the pandemic and its effect on classes. it therefore establishes a degree of plausibility of the relationships depicted in the model. Hence our call for replication studies.

The model used to guide this research provides a useful framework for studying other forms of support and compatibility. Further, our model may be applied to other technology-mediated, distant forms of work, such as telework and virtual teams, although some modifications may be necessary such as substituting managerial support for faculty support. Another way our model may be applied to future research is to consider other forms of support, such as social support or peer support. Other aspects of compatibility such as compatibility with values or compatibility with experience may also be interesting additions. We strongly suggest that researchers carefully consider their research context when studying other forms of organizational support and compatibility; context will affect which forms of support and compatibility are salient in a particular circumstance.

We recommended applying the model to future studies that also focus on the difference in class sizes. With smaller classes, there might be an inclination towards distance learning, as

this study focused predominantly on courses with large class sizes at the two institutions (e.g., some courses have more than 500 students in a group).

Lastly, we recommend a comparison study based on infrastructure. In this study, students can access good campus infrastructure. However, at home, the picture is entirely different. It will be interesting to see the outcomes in a first-world country, where the home infrastructure is also as good, and even better, than on campus.

## Supporting information

**S1 Appendix. Scale items and sources** [8, 16, 46, 65–67].
(DOCX)

**S1 File.**
(CSV)

## Author Contributions

**Conceptualization:** Craig van Slyke, Geoffrey Dick.

**Data curation:** Craig van Slyke.

**Formal analysis:** Craig van Slyke, Geoffrey Dick, Hossana Twinomurinzi, Lateef Babatunde Amusa.

**Investigation:** Adriana Aletta Steyn, Geoffrey Dick.

**Methodology:** Craig van Slyke, Geoffrey Dick, Hossana Twinomurinzi.

**Project administration:** Adriana Aletta Steyn.

**Resources:** Adriana Aletta Steyn.

**Validation:** Adriana Aletta Steyn.

**Writing – original draft:** Adriana Aletta Steyn.

**Writing – review & editing:** Adriana Aletta Steyn, Craig van Slyke, Geoffrey Dick, Hossana Twinomurinzi, Lateef Babatunde Amusa.

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
