## [Decision Letter · Decision Letter 0]

9 May 2023

PONE-D-23-06971Student intentions to continue with distance learning post-COVID: An empirical analysisPLOS ONE

Dear Dr. Steyn,

Thank you for submitting your manuscript to PLOS ONE. After careful consideration, we feel that it has merit but does not fully meet PLOS ONE’s publication criteria as it currently stands. Therefore, we invite you to submit a revised version of the manuscript that addresses the points raised during the review process.

We look forward to receiving your revised manuscript.

Kind regards,

Mohammed A. Al-Sharafi

Academic Editor

PLOS ONE

4. Please remove your figures from within your manuscript file, leaving only the individual TIFF/EPS image files, uploaded separately. These will be automatically included in the reviewers’ PDF.

Additional Editor Comments:

We have received feedback from three reviewers. We appreciate the effort you have put into this research, and we would like to inform you that the reviewers have provided constructive suggestions for improvements. Please address each comment one by one and highlight the revisions made in your manuscript. Here is a summary of their suggestions:

1. Improve the abstract, results section, and discussion of findings.

2. Strengthen the introduction, methodology, and discussion sections.

3. Enhance the literature review and theoretical support.

4. Include more studies related to student intentions to continue with distance learning.

5. Assess and address common method bias.

6. Address issues with HTMT values, research model, and measurement items.

7. Revise the implications section to provide specific and actionable suggestions.

8. Elaborate on the significance of the study in the conclusion section.

9. Address limitations and future work.

10. Proofread the manuscript for typos and mistakes.

When submitting your revised manuscript, please provide a point-by-point response detailing how you have addressed each suggestion. This will help the reviewers and me better understand the changes you have made and expedite the review process.

Reviewers' comments:

Reviewer's Responses to Questions

**Comments to the Author**

1. Is the manuscript technically sound, and do the data support the conclusions?

Reviewer #1: Yes

Reviewer #2: Yes

Reviewer #3: Yes

2. Has the statistical analysis been performed appropriately and rigorously? 

Reviewer #1: Yes

Reviewer #2: Yes

Reviewer #3: Yes

3. Have the authors made all data underlying the findings in their manuscript fully available?

Reviewer #1: Yes

Reviewer #2: No

Reviewer #3: Yes

4. Is the manuscript presented in an intelligible fashion and written in standard English?

Reviewer #1: Yes

Reviewer #2: Yes

Reviewer #3: Yes

5. Review Comments to the Author

Reviewer #1: Thank you for submitting you research paper for PLOS ONE. There are some aspects that need to be addressed by the authors in order to increase the quality of this paper.

1. Further the introduction is a bit weak. Why is the research needed now? Look at how to create a compelling problematization and a hook for readers to be engaged in your study. Problematization is extremely important to showcase why your findings are going to be useful for explaining the phenomenon in the future. Explicitly introduce research questions / research objectives towards the end of the introduction, as bullet points, before sharing how the remaining article is structured.

2.The research methodology needs strengthening. Can you introduce subsections for data collection and data analysis methodologies separately? What could have been trade-offs or biases in your study? Particularly, define the population of the study and sampling technique adopted. Also please justify why your chosen sampling technique and sample size are appropriate.

3. The discussion part can be broadened a bit more with more relevant research studies, as we know there are huge volume of research in this area.

4. I found the implications section to be somewhat weak, as there is a lack of clear connection between the results of the quantitative data analysis and practical implications. The authors should consider revising the implications section to provide more specific and actionable suggestions for practitioners, developers, and designers of metaverse-based learning platforms in higher education.

5. The literature review, the theoretical support, and the discussion sections can benefit from well-established related research, including:

-Developing a holistic success model for sustainable e-learning: A structural equation modeling approach. https://doi.org/10.3390/su13169453

- Novel extension of the UTAUT model to understand continued usage intention of learning management systems: The role of learning tradition. https://doi.org/10.1007/s10639-021-10758-y

- Exploring Student Readiness to MOOCs in Jordan: A Structural Equation Modelling Approach. https://doi.org/10.28945/4542

- Towards a Sustainable Adoption of E-Learning Systems: The Role of Self-Directed Learning. https://doi.org/10.28945/4980

Reviewer #2: Full Title: Student intentions to continue with distance learning post-COVID: An empirical analysis

Objective: The present study aimed to investigate investigated the organizational and individual factors that influence the preference for continuing with the distance/online learning post-COVID.

The manuscript needs a proof reading. The author(s) should check English since there are (much too) many instances of typos and mistakes throughout the manuscript.

The authors claims that “All the HTMT values were lower than the conservative threshold of 0.85.” However, the HTMT value between 2 and 8 in Table 3 is 0.867.

Figure 2 provides a research model showing the relationships between study variables. Please use a dashed line to indicate nonsignificant paths.

The items measuring each dimension in the model were provided. Several items are not relevant to the study context (distance learning). For example, “My professors really care about me; I can rely on my professors; My school really cares about my well-being; My school is concerned about me as a person.” Each item should be related to distance learning.

Further, as the hypotheses were related to the study context “e.g., the perceived institutional support for distance learning will positively influence students’ satisfaction with distance learning” each measurement item should be tailored to the study context.

Descriptive Statistics (Mean, S.D., Skewness, Kurtosis) for each subdimension should be reported.

Since this is a cross-sectional study, common method bias may arise from certain tendencies that respondents apply. Therefore, common method bias should be an important concern and assessed.

Discussion & Conclusion should be enlarged, including a better interpretation of results and a better justification of the conclusions. Refer to the following relevant articles in the revised manuscript: “Generation Z use of artificial intelligence products and its impact on environmental sustainability: A cross-cultural comparison”, “Examining the impact of psychological, social, and quality factors on the continuous intention to use virtual meeting platforms during and beyond COVID-19 pandemic: A hybrid SEM-ANN approach”, “The impact of knowledge management practices on the acceptance of Massive Open Online Courses (MOOCs) by engineering students: A cross-cultural comparison”, “The role of self-efficacy in predicting use of distance education tools and learning management systems”, and “The role of self-directed learning, metacognition, and 21st century skills predicting the readiness for online learning.”

Reviewer #3: Student intentions to continue with distance learning post-COVID: An empirical analysis

Below are comments to improve the manuscript

The abstract does not discuss on data collection and data analysis tools. The key findings are not well written. Thus, improve the main issue to be addressed in the study and finding part of the abstract.

The introduction section needs to be improved to include the main research problems and research questions.

Included studies related to Student intentions to continue with distance learning such as;

Alarabiat, A., Hujran, O., Soares, D., & Tarhini, A. (2023). Examining students' continuous use of online learning in the post-COVID-19 era: an application of the process virtualization theory. Information Technology & People, 36(1), 21-47.

Anthony Jnr, B., & Noel, S. (2021). Examining the adoption of emergency remote teaching and virtual learning during and after COVID-19 pandemic. International Journal of Educational Management, 35(6), 1136-1150.

Marandu, E. E., Mathew, I. R., Svotwa, T. D., Machera, R. P., & Jaiyeoba, O. (2023). Predicting students' intention to continue online learning post-COVID-19 pandemic: extension of the unified theory of acceptance and usage technology. Journal of Applied Research in Higher Education, 15(3), 681-697.

Jnr, B. A., Kamaludin, A., Romli, A., Raffei, A. F. M., Phon, D. N. A. E., Abdullah, A., ... & Baba, S. (2020). Predictors of blended learning deployment in institutions of higher learning: theory of planned behavior perspective. The International Journal of Information and Learning Technology, 37(4), 179-196.

What sampling methods was employed in the study. Also, provide justification on the employed methodology (statistical tool if any e.g) utilized in the study.

How was the reliability and validity assessed for this study. This should be well discussed based on existing scales or benchmark from the literature.

Improve the results section and related back to the literature. What are the main discoveries from the study, can you summarize as a model (figure) or table

I miss a comparison of your results as compared to other studies in the literature on Student intentions to continue with distance learning.

The discussion section needs to discuss findings from the literature and also findings from the current study.

Also, include the research implications of the study as a different section. This can be implication for theory as well as the implication for practice/policy

The conclusion section should elaborate more on the significance of the study.

Lastly improve the limitations and future works.

In particular, what is the main contribution from this study it’s hard to see

OVERALL, write manuscript as introduction, literature review, methodology, findings, discussion and implications, conclusion.

Good luck

6. PLOS authors have the option to publish the peer review history of their article (what does this mean?). If published, this will include your full peer review and any attached files.

Reviewer #1: **Yes: **Ahmad Samed Al-Adwan

Reviewer #2: No

Reviewer #3: No

---

## [Author Response · Author response to Decision Letter 0]

14 Aug 2023

Thank you to all the reviewers for a very comprehensive and thorough review. We have attempted to address each of them, as detailed in the reviewer's response document submitted.

---

## [Decision Letter · Decision Letter 1]

5 Oct 2023

Student intentions to continue with distance learning post-COVID: An empirical analysis

PONE-D-23-06971R1

Dear Dr. Steyn,

We’re pleased to inform you that your manuscript has been judged scientifically suitable for publication and will be formally accepted for publication once it meets all outstanding technical requirements.

Kind regards,

Mohammed A. Al-Sharafi

Academic Editor

PLOS ONE

Reviewers' comments:

Reviewer's Responses to Questions

**Comments to the Author**

1. If the authors have adequately addressed your comments raised in a previous round of review and you feel that this manuscript is now acceptable for publication, you may indicate that here to bypass the “Comments to the Author” section, enter your conflict of interest statement in the “Confidential to Editor” section, and submit your "Accept" recommendation.

Reviewer #1: All comments have been addressed

2. Is the manuscript technically sound, and do the data support the conclusions?

Reviewer #1: Yes

3. Has the statistical analysis been performed appropriately and rigorously? 

Reviewer #1: Yes

4. Have the authors made all data underlying the findings in their manuscript fully available?

Reviewer #1: Yes

5. Is the manuscript presented in an intelligible fashion and written in standard English?

Reviewer #1: Yes

6. Review Comments to the Author

Reviewer #1: Thank you for submitting the revised version of your paper. You have addressed most of the reviewers' comments. Great job. Consequently, the quality of this submission has improved significantly.

7. PLOS authors have the option to publish the peer review history of their article (what does this mean?). If published, this will include your full peer review and any attached files.

Reviewer #1: No

---

## [Editor Report · Acceptance letter]

13 Oct 2023

PONE-D-23-06971R1 

Student intentions to continue with distance learning post-COVID: An empirical analysis 

Dear Dr. Steyn:

I'm pleased to inform you that your manuscript has been deemed suitable for publication in PLOS ONE. Congratulations! Your manuscript is now with our production department. 

Kind regards, 

on behalf of

Dr. Mohammed A. Al-Sharafi 

Academic Editor

PLOS ONE